# Influence of Pharmaceutical Copayment on Emergency Hospital Admissions: A 1978–2018 Time Series Analysis in Spain

**DOI:** 10.3390/ijerph18158009

**Published:** 2021-07-29

**Authors:** Antonio Palazón-Bru, Miriam Calvo-Pérez, Pilar Rico-Ferreira, María Anunciación Freire-Ballesta, Vicente Francisco Gil-Guillén, María de los Ángeles Carbonell-Torregrosa

**Affiliations:** 1Department of Clinical Medicine, Miguel Hernández University, 03550 Alicante, Spain; vte.gil@gmail.com (V.F.G.-G.); carbonell_mar@gva.es (M.d.l.Á.C.-T.); 2Primary Care Pharmacy Service, General University Hospital of Elda, 03600 Alicante, Spain; calvo_mir@gva.es (M.C.-P.); rico_pil@gva.es (P.R.-F.); 3Primary Care Pharmacy Service, General University Hospital of Alicante, 03010 Alicante, Spain; cristina.avilaf@gmail.com; 4Emergency Services, General University Hospital of Elda, 03600 Alicante, Spain

**Keywords:** pharmacy, copayment, health expenditures, cost control, hospitalization, inpatients

## Abstract

No studies have evaluated the influence of pharmaceutical copayment on hospital admission rates using time series analysis. Therefore, we aimed to analyze the relationship between hospital admission rates and the influence of the introduction of a pharmaceutical copayment system (PCS). In July 2012, a PCS was implemented in Spain, and we designed a time series analysis (1978–2018) to assess its impact on emergency hospital admissions. Hospital admission rates were estimated between 1978 and 2018 each month using the Hospital Morbidity Survey in Spain (the number of urgent hospital admissions per 100,000 inhabitants). This was conducted for men, women and both and for all-cause, cardiovascular and respiratory hospital discharges. Life expectancy was obtained from the National Institute of Statistics. The copayment variable took a value of 0 before its implementation (pre-PCS: January 1978–June 2012) and 1 after that (post-PCS: July 2012–December 2018). ARIMA (Autoregressive Integrated Moving Average) (2,0,0)(1,0,0) models were estimated with two predictors (life expectancy and copayment implementation). Pharmaceutical copayment did not influence hospital admission rates (with *p*-values between 0.448 and 0.925) and there was even a reduction in the rates for most of the analyses performed. In conclusion, the PCS did not influence hospital admission rates. More studies are needed to design health policies that strike a balance between the amount contributed by the taxpayer and hospital admission rates.

## 1. Introduction

A chronic disease is a physical or mental condition that lasts more than one year and causes functional limitations or requires ongoing monitoring or treatment [1]. Approximately one person in three has more than one chronic disease [2,3], resulting in high healthcare costs [2,4], attributable to the high utilization of primary and specialty care services, increased medication use, higher frequency of emergency department visits and hospital admissions [5]. Medication is supplied by the health care system and patients, depending on their situation, must make a variable contribution to outpatient pharmaceutical services [6].

Different studies have evaluated the influence of pharmaceutical copayments on different aspects of health [7,8,9,10,11,12,13,14,15,16,17,18], such as adherence to therapy [7,9,10,16], smoking cessation [8], reduction in the use of drugs [11,12,15,17], physician consultations [13,14,18] and hospital admissions [14,16,18]. Regarding hospital admissions [14,16,18], two studies have focused on specific patients [16,18], assessing the impact of either increasing or decreasing pharmaceutical costs during one year. The only significant finding was an increase in hospitalizations due to chronic obstructive pulmonary disease, asthma or emphysema in older long-term users of inhaled medications [18]. However, the study data were collected between 2001 and 2004, and this could have changed over time. Another study analyzed two cross-sections before and after the implementation of a copayment system [14], both in Spain and Germany, assessing hospitalizations in the previous year, with data obtained through surveys. The authors of this paper did not compare both periods (pre and post-intervention) and only provided the percentages of hospitalizations, classified by socioeconomic level in the two countries, which generally showed a reduction after the implementation of the copayment system [14].

The scientific literature does not appear to have addressed in detail whether the implementation of a pharmaceutical copayment system (PCS) influences the hospitalization rate, since it has only been evaluated with very specific patients and/or with a brief or cross-sectional follow-up. We need to know whether, at the population level and in the long term, this implementation produces any change in the hospitalization rate, taking into account the seasonality of admissions and the increase in the life expectancy of the population—key factors when analyzing this relevant issue. Patients may not have been able to acquire their medicines and would therefore have failed to adhere to their treatment [11,15,19,20], which could result in a greater incidence of hospital admissions among chronic patients [21,22,23]. Thus, if this hypothetical relation existed, there could be a certain difference in the rates of hospital admissions between the periods before and after the start of the PCS. This possibility is worth studying given that it is a relevant public health question that has not, as far as we are aware, yet been examined. Consequently, we conducted a study in Spain using population data between 1978 and 2018, postulating that the pharmaceutical copayment implemented in July 2012 has not influenced hospital admissions rates. Our research objectives were thus to determine whether the PCS influenced hospital admission rates through a time series analysis. We selected Spain, a developed country with a free health care system, which could then serve as a basis for further studies in developing countries where the burden for medical expenses is shouldered by patients or co-payers.

## 2. Materials and Methods

### 2.1. Setting, Study Population, Study Design and Participants

In Spain, medical care (primary, specialized and hospital care) and emergency room visits are completely free and universal. Patient contributions to pharmaceutical services until June 2012 were based on the Royal Decrees 945/1978, 1605/1980 and 83/1993 [24,25,26]. These decrees stated that active workers were required to make a contribution of 40%, those suffering from HIV and chronically ill patients would contribute 10%, and pensioners, disabled individuals and those with work-related illnesses were exempt. Subsequently, the Royal Decree-Law 16/2012 and Royal Decree 1192/2012 and the Resolution of 15 December 2014 came into force whereby changes were established in patient contribution to pharmaceutical services, which have been in effect since 1 January 2015 [27,28,29]. These changes are shown in Table 1 [27,28,29].

Definitions: (1) Unemployment subsidy: economic aid received by unemployed persons who are not entitled to unemployment benefits, aimed at individuals and collectives without economic resources and who find it difficult to integrate; (2) Unemployment benefit: contributory benefit for loss of work, the duration and amount of which is determined by the length of time the worker has paid unemployment contributions into the Social Security system; (3) Noncontributory pensions: economic benefits for citizens who lack sufficient resources for subsistence, even if they have never contributed or contributed for an insufficient length of time to reach contributory benefits; (4) Social Integration Income: social benefit intended to ensure economic subsistence resources for those who lack them, personal, family, social and, if applicable, labor integration actions; (5) Work accident: an accident suffered by a worker during working hours or on the way to work or from work to home; (6) Occupational disease: disease contracted as a result of work for another party; (7) Toxic oil syndrome: mass poisoning suffered in Spain in the spring of 1981. The disease affected more than 20,000 people, causing the death of about 330 people. In 1989, the Supreme Court considered the causal link between the intake of denatured rapeseed oil and disease to be proven; (8) Disability: a condition under which certain people have a physical, mental, intellectual or sensory impairment that, in the long term, affects the way they interact and participate fully in society.

The study population included all inhabitants in Spain between 1978 and 2018.

This time series study analyzed the evolution of the mean emergency admissions rate per 100,000 inhabitants in each month from January 1978 to December 2018, establishing a cut-off point to determine whether copayment implementation (July 2012) influenced this evolution. The data for constructing the series were obtained from the Hospital Morbidity Survey, which has been carried out in Spain since 1977. The data have been available for download in an anonymized and encrypted form since 1978 [30,31], together with the total Spanish population figures for each year, which the National Institute of Statistics prepares annually [32].

Although the methodology of the survey has been previously published [30,31], we summarize it here. The survey is national in scope, extending to all public and private sector hospitals, as well as military hospitals, given the large contingent of civilian personnel who are treated in them. A patient is considered to be a person who has been admitted to a hospital to be treated, diagnosed or observed as an inpatient and who has been discharged from the hospital. The selection of patients was carried out through systematic sampling. The initial diagnosis is classified as ordinary or emergency, while the main diagnosis corresponds to the cause of admission, according to the criteria of the clinical service or physician who attended the patient. ICD-9-CM coding was used from 1978 to 2015, and ICD-10-CM for subsequent years. The date of discharge is also indicated, which may be due to cure, transfer to another center, death or other causes, together with sex, age, province, days of stay and the sampling factor; i.e., the weight of each patient in the estimates [30,31]. This weight was used to calculate our variables.

### 2.2. Ethical Aspects

The data collected were extracted through the website of the National Institute of Statistics, in a completely anonymized and encrypted form, thus preventing the identification of any individual person [33]. This work was approved by the Ethics Committee of the Miguel Hernández University (reference: AUT.DMC.APB.01.21).

### 2.3. Variables and Measurements, and Statistical Methods

The main study variable was the emergency admissions rate per 100,000 inhabitants for each month and year, starting in January 1978 and ending in December 2018. To calculate this rate, all the weights of patients with a diagnosis of emergency admission and discharge date in the corresponding month and year were summed, dividing this amount by the total population of that year. This was done after separating men and women, changing the population totals accordingly. The mean emergency admissions rate for cardiovascular or respiratory causes was also determined for men, women and both, using the ICD-9-CM and ICD-10-CM codes. The data sources were the Hospital Morbidity Survey [30,31] and the total population figures of the National Institute of Statistics [32].

The copayment indicator variable was defined, taking a value of 1 as of July 2012 and 0 before that date, together with the population life expectancy. A distinction was made between men and women, when necessary, with this information also obtained from the National Statistics Institute (INE) [34].

Each time series (hospital discharges for each month and each year, from January 1978 to December 2018, either all-cause, cardiovascular or respiratory cause; male, female or both) was plotted on a time plot, and the Ljung-Box Q test was applied to ascertain whether the data were independently distributed. As a strong dependence between hospital admissions was found, ARIMA (2,0,0)(1,0,0) models were estimated for each series (12-month seasonality), without transforming the data. The justification for the model parameters lies in the seasonality of admissions, both on an annual and seasonal basis (each season—spring, summer, autumn and winter—has 3 months). This was directly visible in the time plots. Two predictors were added to all the models: pharmaceutical copayment and life expectancy in that year. We wanted to determine whether the former had an influence on admissions, and the latter was a key adjustment factor, since as the life expectancy of the population increases, annual admissions increases as a result. The stationary R-squared was calculated to determine the goodness of fit of the models. No transformations were applied to obtain simple models, and the coefficient associated with the predictors indicated how they directly affect the rate of admissions per 100,000 inhabitants in the population. A step-by-step procedure of the analysis was as follows (with data for our case in parenthesis): (1) identify the dependent variable (rate of admissions per 100,000 inhabitants); (2) select the explanatory variables (life expectancy and PCS); (3) determine whether the data were independently distributed; (4) establish the difference order in the ARIMA, depending on the nature of the data (two in our case, because we are considering the seasons: spring, summer, autumn and winter); and (5) consider whether there is seasonality (in our case, we have annual data). Although several methods exist for modeling time series, we selected ARIMA as it is easy to understand and perfectly reflects the time relation between the data mentioned, particularly when there is seasonality. In the case of hospital admissions, similar rates were found in the different seasons, with similar results between one year and the previous year. In addition, this enabled predictors to be added to see their impact on the dependent variable. This type of model was, therefore, suitable for our purpose. More details about our models are given in Appendix A. All analyses were performed with a significance level of 5%, and for each relevant parameter, its associated confidence interval (CI) was calculated. The statistical package used was IBM SPSS Statistics 26 (Armonk, New York, NY, USA).

## 3. Results

The time graphs are shown in Figure 1, Figure 2 and Figure 3, distinguishing between types of admission (all, cardiovascular and respiratory) and sex (men, women or both). They show a clear seasonality and an increase in admissions over time, which began to stabilize at the beginning of the 21st century. The series data did not show independence (*p* < 0.001 in all cases, Table 2) and the adjusted ARIMA (2,0,0)(1,0,0)(1,0,0) models had an excellent fit, as the stationary R-squared ranged from 0.919 to 0.974—very close to the perfect fit (value 1).

The dotted line indicates when the copayment system was implemented. The top figure corresponds to all-cause admissions, the middle figure to cardiovascular admissions and the bottom figure to respiratory admissions.

The dotted line indicates when the copayment system was implemented. The top figure corresponds to all-cause admissions, the middle figure to cardiovascular admissions and the bottom figure to respiratory admissions.

The dotted line indicates when the copayment system was implemented. The top figure corresponds to all-cause admissions, the middle figure to cardiovascular admissions and the bottom figure to respiratory admissions.

The coefficients of the models are presented in Table 3 and show that all the coefficients related to seasonality were highly significant, together with the life expectancy of the population. The pharmaceutical copayment was not significant in any case and, except for respiratory admissions for women, there were fewer admissions when this health policy was implemented (negative coefficients in the models), and this increase was only 0.7 admissions per 100,000 women (*p* = 0.857).

## 4. Discussion

### 4.1. Summary

The results of this study show that the pharmaceutical copayment system had no influence on the hospital admission rate. This lack of association was equally true for cardiovascular and respiratory admissions, regardless of the sex of the patients. An increase in the rate of hospitalization was observed, but it seemed to be due to the increase in the life expectancy of the population.

### 4.2. Strengths and Limitations

The main strength of this study is its assessment of the influence of a pharmaceutical copayment system on the hospital admission rate through a time series analysis including data from 1978 to 2018. These results are innovative as our literature search found only three papers assessing this association that compared only two time points without a long time series. Clearly, the association between the pharmaceutical copayment system and a change in the hospital admission rate should be evaluated through a randomized intervention study with a control group. However, since the pharmaceutical copayment system was implemented across the country at the same time, the design used was the best option to address our research question. Although we could have selected a shorter time period for the study nearer to the implementation of the PCS, we preferred to study the time series available with 40 years of data with precision. Logically, there are great differences in the early stages, but by considering life expectancy as a covariable, the ARIMA models have a very good fit, showing stationary R-squared values near 1, indicating that the choice of models was very suitable. We could also have used another type of econometric model, such as linear regression. However, we must bear in mind that there exists a temporality and we were not dealing with independent data. Thus, for this and the other reasons mentioned, we preferred to use the ARIMA models, which adjusted very well to our data and helped to answer our research question.

Regarding selection bias, we must bear in mind that the National Institute of Statistics selects all public and private hospitals in the national territory for sampling, thus minimizing this bias [30,31]. Concerning information bias, in the design of the Hospital Morbidity Survey, it is indicated that data quality criteria are applied when obtaining the information, as well as strict validation and coherence controls between the variables; thus, the precision and reliability of the data are quite high [33]. In addition, the sampling weight of each patient has been reduced over time, which means that in recent years, practically the entire population was sampled. The mean statistical weight in 1978 was 12.48, which was reduced to less than 10 from 1988, to less than 5 from 1994, to 2 from 1998 and to less than 1.1 from 2006. Finally, time series models with a high adjustment capacity were estimated, taking into account the seasonality of hospital admissions, together with the increase in the life expectancy of the population, which minimizes possible confounding biases.

### 4.3. Comparison with the Existing Literature

When comparing our results with those published by other authors, we encountered a difficulty in that two of the three relevant studies found were conducted in public health systems that are very different from the Spanish public health system [16,18]. Furthermore, these two studies focused on very specific patients, and the intervention targeted a sample and not the entire population, as in our case. In contrast, the other study consulted [14] was carried out in Spain and evaluated the same copayment system. However, this study only determined the percentage of individuals who had been admitted the previous year at two time points, before and after the implementation of the copayment, without performing any type of statistical comparison between these time periods. Consequently, it appears that our study is the first to provide results at the population level on the influence of the copayment system on the hospital admission rate, taking into account the seasonality of admissions and an analysis of more than 40 years of hospital discharges.

Although the association between the PCS and hospital admissions was not addressed in depth, we can nevertheless suggest a hypothesis about how it would work for hospital admissions in other types of health care systems, as found for example in developing countries with a lack of free public health care or countries dominated by private health insurance. The former would represent a situation in which most of the population have limited resources and the latter a situation in which citizens without sufficient economic resources would be unable to acquire medical insurance with good cover. We consider that a PCS would have a great impact in the latter case as, if the state is able to subsidize part of the pharmacological costs, the insurance companies could then reduce the individual quotas payable, thereby enabling greater health care cover to be acquired, which in turn could result in fewer hospital admissions. On the other hand, if used in developing countries, it would alleviate expenses, resulting in improved individual health [35]. Another situation that could change hospital admission rates is the COVID-19 pandemic, which has caused great changes around the world, with many citizens afraid to go to a hospital in case they become infected [36]. Nevertheless, all these issues need to be assessed in future studies, as the impact of the pandemic has not yet been fully evaluated concerning such aspects as diagnostic delays, changes in the childhood vaccination calendars, treatment changes, etc., which could all have an impact on future hospital admission rates. 

To attempt to understand why the results were not statistically significant, we should analyze Table 1 and the population demographics. This shows that the most vulnerable persons, such as pensioners, the long-term unemployed who have lost their subsidy and the disabled, are exempt from copayment. However, copayment can cause difficulties regarding access to drugs for those who are worse-off financially, such as low-income workers. Considering that those most likely to be admitted are the elderly, this could explain the lack of an association between the PCS and the greater rate of hospital admissions. Nonetheless, we have to be cautious and design more accurate studies to determine whether the causes of admission are more associated with the lack of adherence to medication and whether this is in fact due to the high cost of the drugs themselves. Independently of these studies, preventive measures can be applied to promote healthy lifestyle habits to thereby minimize the possibility of hospital admission.

### 4.4. Implications to Public Health and Research

Considering that we have found no studies assessing the influence of pharmaceutical copayment on hospital admission rates with time series, it would of interest to conduct studies in other health care systems. This is an important issue because there are health systems with fixed or variable fees according to prescription, drug classification by disease or effectiveness, etc. [6]. This variability in pharmaceutical services systems could generate differences in hospital admission rates. In other words, more studies evaluating this association are needed to provide further scientific evidence and to design health policies to benefit the patient.

## 5. Conclusions

In a country with a free health care system in which a pharmaceutical services copayment system was introduced, the new system did not influence the hospital admission rate, not even for cardiovascular and respiratory diseases or when distinguishing between male and female patients. As a conclusion, in light of these results, we need more studies to be able to propose health policies that strike a balance between the amount contributed by the taxpayer and hospital admission rates. However, this should be without prejudice to the health of the citizens and should attempt to reduce hospital admissions with preventive policies such as encouraging healthy lifestyle habits. This study also shows that, given the great differences between health care systems, it is extremely important to try to examine our research question elsewhere so that we can determine whether this lack of association between PCS and hospital admissions is also found in a system that is unlike the Spanish system (free and with free access to all health services except for medicines, for which copayment has existed since 2012).

## Figures and Tables

**Figure 1 ijerph-18-08009-f001:**
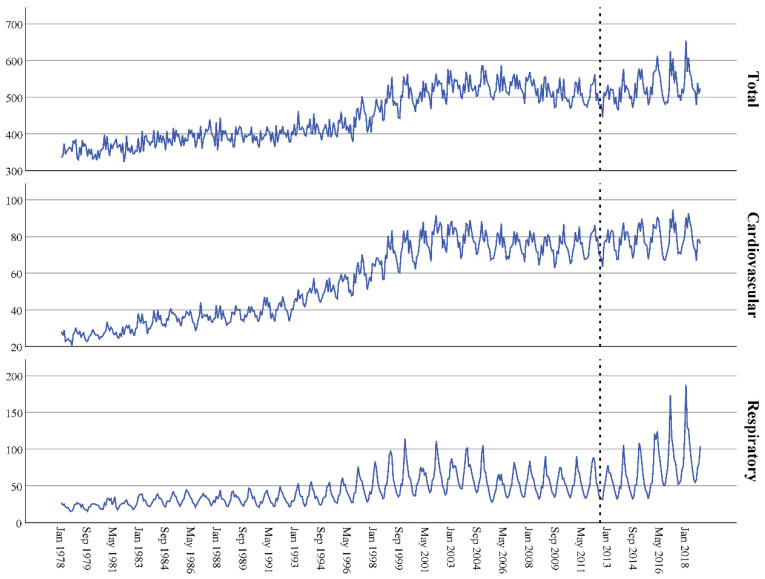
Time series for the number of hospital admissions per 100,000 inhabitants in Spain: 1978–2018 data.

**Figure 2 ijerph-18-08009-f002:**
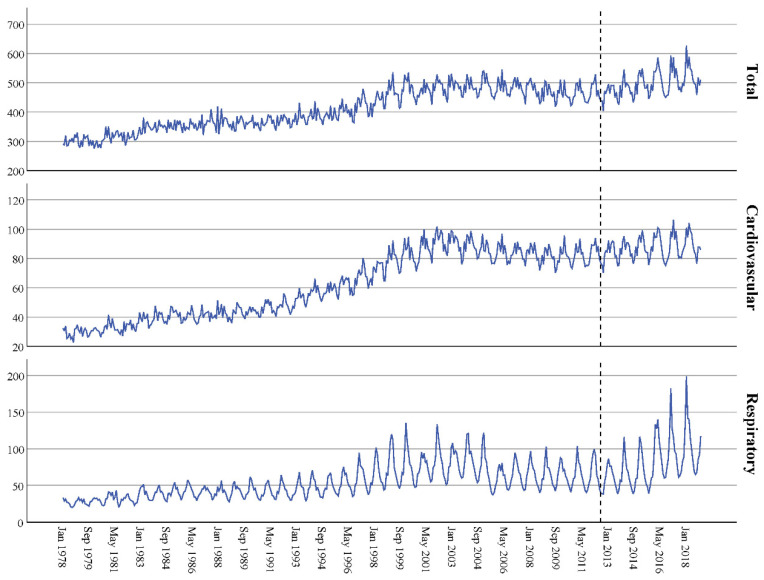
Time series for the number of hospital admissions per 100,000 male inhabitants in Spain: 1978–2018 data.

**Figure 3 ijerph-18-08009-f003:**
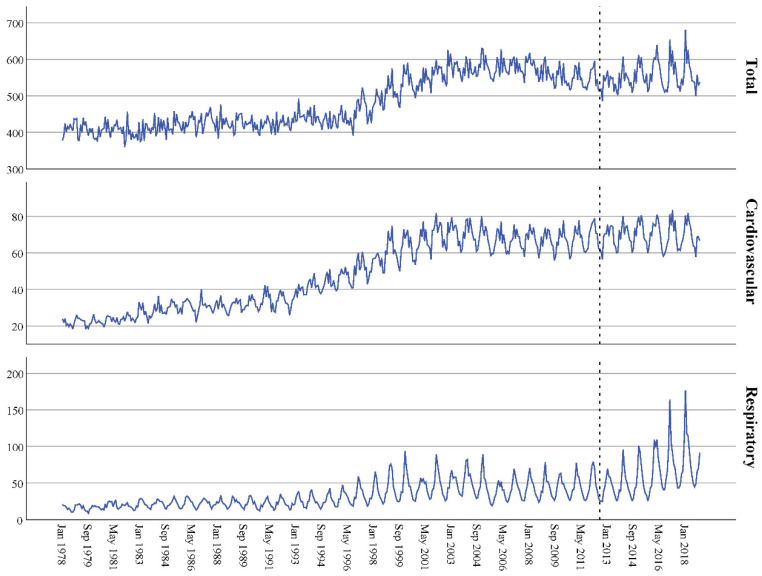
Time series for the number of hospital admissions per 100,000 female inhabitants in Spain: 1978–2018 data.

**Table 1 ijerph-18-08009-t001:** Patient contribution to pharmaceutical services after implementation of the pharmaceutical copayment (Royal Decree Law 16/2012 and Royal Decree 1192/2012).

Category	Contribution (%)
Exempt from payment: Unemployed individuals not receiving unemployment benefits Recipients of noncontributory pensions Recipients of social integration income Treatments resulting from work-related accidents and occupational diseases Individuals affected by toxic oil syndrome and people with disabilities	0
Pensioners (income in €):	
<18,000	10 (up to 8.23 €/month)
18,000–99,999	10 (up to 18.52 €/month)
≥100,000	60 (up to 61.75 €/month)
Assets (income in €):	
<18,000	40
18,000–99,999	50
≥100,000	60

**Table 2 ijerph-18-08009-t002:** Goodness of fit of the ARIMA (2,0,0)(1,0,0) models to assess the impact of copayment on the rate of emergency hospital admissions per 100,000 inhabitants in Spain: 1978–2018 data.

Parameter	Men			Women			Both		
	Total	Cardiovascular	Respiratory	Total	Cardiovascular	Respiratory	Total	Cardiovascular	Respiratory
Stationary R-squared	0.930	0.969	0.919	0.928	0.970	0.919	0.933	0.974	0.922
Ljung-Box Q (*p*-value)	<0.001	<0.001	<0.001	<0.001	<0.001	<0.001	<0.001	<0.001	<0.001

**Table 3 ijerph-18-08009-t003:** Coefficients of the ARIMA (2,0,0)(1,0,0) models to assess the impact of copayment on the rate of emergency hospital admissions per 100,000 inhabitants in Spain: 1978–2018 data.

*Variable*	Men						Women						Both					
	Total		Cardiovascular		Respiratory		Total		Cardiovascular		Respiratory		Total		Cardiovascular		Respiratory	
	B ± SE	*p*	B ± SE	*p*	B ± SE	*p*	B ± SE	*p*	B ± SE	*p*	B ± SE	*p*	B ± SE	*p*	B ± SE	*p*	B ± SE	*p*
Constant	−1138 ± 193	<0.001	−417 ± 52	<0.001	−373 ± 100	<0.001	−995 ± 251	<0.001	−378 ± 49	<0.001	−293 ± 98	0.003	−1115 ± 209	<0.001	−379 ± 52	<0.001	−350 ± 98	<0.001
AR:																		
Lag 1	0.35 ± 0.04	<0.001	0.40 ± 0.04	<0.001	0.80 ± 0.05	<0.001	0.43 ± 0.04	<0.001	0.37 ± 0.05	<0.001	0.77 ± 0.05	<0.001	0.38 ± 0.04	<0.001	0.38 ± 0.04	<0.001	0.81 ± 0.04	<0.001
Lag 2	0.25 ± 0.04	<0.001	0.32 ± 0.04	<0.001	−0.15 ± 0.04	<0.001	0.20 ± 0.04	<0.001	0.30 ± 0.04	<0.001	−0.18 ± 0.04	<0.001	0.21 ± 0.04	<0.001	0.33 ± 0.04	<0.001	−0.19 ± 0.04	<0.001
AR (seasonal):																		
Lag 1	0.72 ± 0.03	<0.001	0.72 ± 0.03	<0.001	0.80 ± 0.03	<0.001	0.74 ± 0.03	<0.001	0.75 ± 0.03	<0.001	0.85 ± 0.03	<0.001	0.74 ± 0.03	<0.001	0.77 ± 0.3	<0.001	0.83 ± 0.03	<0.001
Copayment	−6.4 ± 11.1	0.565	−2.0 ± 2.6	0.448	−0.5 ± 4.8	0.925	−6.2 ± 11.5	0.589	0.5 ± 2.0	0.792	1.8 ± 3.7	0.624	−7.2 ± 10.7	0.503	−0.3 ± 2.1	0.900	0.7 ± 4.1	0.857
Life expectancy	20.6 ± 2.6	<0.001	6.3 ± 0.7	<0.001	5.7 ± 1.3	<0.001	18.1 ± 3.1	<0.001	5.2 ± 0.6	<0.001	4.1 ± 1.2	<0.001	19.9 ± 2.7	<0.001	5.5 ± 0.7	<0.001	5.1 ± 1.2	<0.001

Abbreviations: B, coefficient; SE, standard error. All the results were adjusted for life expectancy.

## Data Availability

Not applicable.

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
