# Peer review of "Influence of Pharmaceutical Copayment on Emergency Hospital Admissions: A 1978–2018 Time Series Analysis in Spain"

_ijerph, 2021, doi:10.3390/ijerph18158009_

Round 1
Reviewer 1 Report
The topic itself is very interesting, but the authorss' introduction on some key issues such as research significance and research methods is not clear.
1. In the “Introduction”, the authors did not clearly introduce the significance of studying the issue of "influence of pharmaceutical copayment on emergency hospital admissions"?
2. Based on the results of the data analysis, the authors did not discuss why PCS had no effect on the emergency hospital admissions. What are the suggestions for practical work? If the authors didn't discuss these points, we can't understand why the authors want to study the influence of PCS on emergency hospital admissions. The significance of this paper is not clear from beginning to end.
3. There are many time series models. Why do the authors use ARIMA model and what are its advantages? In addition, the authors did not explain why ARIMA (2,0,0) (1,0,0) model was used? The research method and data analysis process are relatively simple.
4. The purpose of this paper was to analyze the influence of pharmaceutical copayment on emergency hospital admissions. PCS has been implemented since July 2012. Is it more appropriate to select the years around 2012 for time series analysis? Why did the authors choose such a long time span from 1978 to 2018? After all, there is a significant difference between the early number of hospital admissions and the current number of hospital admissions.
Author Response
The topic itself is very interesting, but the authorss' introduction on some key issues such as research significance and research methods is not clear.
Thank you very much for your feedback. We have followed your suggestions so as to improve our manuscript.
- In the “Introduction”, the authors did not clearly introduce the significance of studying the issue of "influence of pharmaceutical copayment on emergency hospital admissions"?
We have provided more information about this at the end of the Introduction.
- Based on the results of the data analysis, the authors did not discuss why PCS had no effect on the emergency hospital admissions. What are the suggestions for practical work? If the authors didn't discuss these points, we can't understand why the authors want to study the influence of PCS on emergency hospital admissions. The significance of this paper is not clear from beginning to end.
We have added new points to address these issues in Comparison with existing literature.
- There are many time series models. Why do the authors use ARIMA model and what are its advantages? In addition, the authors did not explain why ARIMA (2,0,0) (1,0,0) model was used? The research method and data analysis process are relatively simple.
We have included new information about why we selected ARIMA models in the last paragraph of Variables and measurements, and statistical methods. On the other hand, in the previous version of the text, we clarified the rationale to use ARIMA (2,0,0) (1,0,0) models: “The justification for the model parameters lies in the seasonality of admissions, both on an annual and seasonal basis (each season has 3 months: spring, summer, autumn and winter). This was directly visible in the time plots.”
- The purpose of this paper was to analyze the influence of pharmaceutical copayment on emergency hospital admissions. PCS has been implemented since July 2012. Is it more appropriate to select the years around 2012 for time series analysis? Why did the authors choose such a long time span from 1978 to 2018? After all, there is a significant difference between the early number of hospital admissions and the current number of hospital admissions.
We have included a comment about this issue in Strengths and limitations.

Reviewer 2 Report
The paper analyzed the relationship between hospital admission rates and the influence of the introduction of a pharmaceutical copayment system in Spain. Applying Auto Regressive Integrated Moving Average (ARIMA) from the time series data, the study found that the pharmaceutical copayment system had no influence on the hospital admission rate which was equally true for cardiovascular and respiratory admissions. This result is a common case for developed countries with free health care system such as Spain, which serves as a basis for further studies in developing countries where the burden for medical expenses are shouldered by the patients (and copayors).
The paper in general is well-written. The structure, objectives, literature gap, academic contribution, and presentation of results are clear and straightforward. Improvements in the Methodology and Discussion may increase the relevance (given its "no influence" result) and significance of the paper. The following are the specific comments to improve the paper.
- Define the research objectives in the Introduction.
- Explain why Spain was selected as a case study which fitted with the research objectives and/or the model.
- Why was time series analysis, specifically ARIMA, chosen over other econometric (statistical) analyses? What are the advantages (and disadvantages)?
- If possible, present the (time series or ARIMA) model in the Methodology identifying the dependent and explanatory variables, standardized residuals, difference order, etc.
- Add a step-by-step procedure of the analysis so that future studies may replicate the Methodology.
- The paper's relevance is in question as the result is very obvious and predictable: copayment has no influence in hospital admission for countries with free public health care system. Therefore, the Discussion must be improved significantly. Some good points for discussion: how is the case for (a) developing countries with lack of free public health care system; (b) countries dominated by private health insurance; (c) pandemic and post-pandemic periods with down health care systems.
- The Conclusion should be increased outlining summary of the study, the take-home message, and the lessons learned.
- Define acronyms on its first use, e.g. ARIMA.
Author Response
The paper analyzed the relationship between hospital admission rates and the influence of the introduction of a pharmaceutical copayment system in Spain. Applying Auto Regressive Integrated Moving Average (ARIMA) from the time series data, the study found that the pharmaceutical copayment system had no influence on the hospital admission rate which was equally true for cardiovascular and respiratory admissions. This result is a common case for developed countries with free health care system such as Spain, which serves as a basis for further studies in developing countries where the burden for medical expenses are shouldered by the patients (and copayors).
The paper in general is well-written. The structure, objectives, literature gap, academic contribution, and presentation of results are clear and straightforward. Improvements in the Methodology and Discussion may increase the relevance (given its "no influence" result) and significance of the paper. The following are the specific comments to improve the paper.
Thank you very much for the positive feedback and the suggestions!
Define the research objectives in the Introduction.
We have edited the last paragraph of the introduction to clarify our objectives.
Explain why Spain was selected as a case study which fitted with the research objectives and/or the model.
We have included more information in the Introduction to answer this question.
Why was time series analysis, specifically ARIMA, chosen over other econometric (statistical) analyses? What are the advantages (and disadvantages)?
A comment about this point has been included in Strengths and limitations.
If possible, present the (time series or ARIMA) model in the Methodology identifying the dependent and explanatory variables, standardized residuals, difference order, etc.
Add a step-by-step procedure of the analysis so that future studies may replicate the Methodology.
Although this had been given in a non-technical way, we have clarified how to estimate the models in more detail. This has been included in the last paragraph of Variables and measurements, and statistical methods.
The paper's relevance is in question as the result is very obvious and predictable: copayment has no influence in hospital admission for countries with free public health care system. Therefore, the Discussion must be improved significantly. Some good points for discussion: how is the case for (a) developing countries with lack of free public health care system; (b) countries dominated by private health insurance; (c) pandemic and post-pandemic periods with down health care systems.
These points have been included in the Discussion, adding new sentences throughout the section.
The Conclusion should be increased outlining summary of the study, the take-home message, and the lessons learned.
We have extended the Conclusion, following your suggestions.
Define acronyms on its first use, e.g. ARIMA.
We apologize for the mistake. We have now defined this acronym at its first mention.

Round 2
Reviewer 1 Report
In the revised version, the authors made a systematic revision and basically answered the relevant questions I raised before. In my opinion, the only deficiency may be that there is still insufficient introduction to the significance of PCs research.
I think the paper meets the requirements for publication.
Author Response
In the revised version, the authors made a systematic revision and basically answered the relevant questions I raised before. In my opinion, the only deficiency may be that there is still insufficient introduction to the significance of PCs research.
As in the previous version of the text the rationale for our study was based on theories. We have, though, included several references to strengthen the statements indicating why the study is clinically relevant. Furthermore, we have included a new phrase to highlight the significance of our research in the introduction.
I think the paper meets the requirements for publication.
Thank you very much for your feedback!

Reviewer 2 Report
The authors made significant changes to improve the manuscript. Most of the reviewer's comments were addressed carefully.
While the paper is non-technical, it is acceptable for medical/health science readers to read the paper as is.
However, readers who have inclination in econometrics and statistics may want to look on the processes behind the results. Specifically, these include the validity of the method used, tests for time series (e.g. significance, stationarity, model identification, diagnostic, residual analysis, forecasting, identifying the difference, and so on. The title with "time series analysis" suggests some technical content in the paper.
Therefore, either put the model in the Methodology OR make it as an Appendix.
Author Response
The authors made significant changes to improve the manuscript. Most of the reviewer's comments were addressed carefully.
Thank you very much for all your feedback, both in this version and for the former version of the text.
While the paper is non-technical, it is acceptable for medical/health science readers to read the paper as is.
However, readers who have inclination in econometrics and statistics may want to look on the processes behind the results. Specifically, these include the validity of the method used, tests for time series (e.g. significance, stationarity, model identification, diagnostic, residual analysis, forecasting, identifying the difference, and so on. The title with "time series analysis" suggests some technical content in the paper.
Therefore, either put the model in the Methodology OR make it as an Appendix.
We have included more details about the statistical model in the Appendix.
